# The Effect of Platelet-Rich Plasma Injection on Short Term Vocal Outcomes Following Phonosurgery—A Pilot Study

**DOI:** 10.3390/medicina58080988

**Published:** 2022-07-25

**Authors:** Laszlo Peter Ujvary, Magdalena Chirilă, Cristina Țiple, Alma Aurelia Maniu, Septimiu Sever Pop, Cristina Maria Blebea, Stefan Vesa, Marcel Cosgarea

**Affiliations:** 1Department of Otolaryngology, “Iuliu Hatieganu” University of medicine and Pharmacy, Cluj, 400349 Cluj-Napoca, Romania; ujvarypeter@outlook.com (L.P.U.); cristinatiple@yahoo.com (C.Ț.); almacjro@yahoo.com (A.A.M.); severpop@me.com (S.S.P.); cristina_blebea@yahoo.com (C.M.B.); rcosgarea@yahoo.com (M.C.); 2Cluj County Emergency Hospital, Cluj, 400006 Cluj-Napoca, Romania; 3Department of Pharmacology, Toxicology and Clinical Pharmacology, “Iuliu Haţieganu” University of Medicine and Pharmacy, 400006 Cluj-Napoca, Romania; stefanvesa@gmail.com

**Keywords:** platelet-rich plasma, PRP, voice analysis, phonosurgery, VHI, voice, vocal fold lesion, benign lesion

## Abstract

*Background and Objectives:* The efficiency and optimal voice rest period following phonosurgery remains debatable. Platelet-rich plasma (PRP) is a safe and cheap alternative to many bioactive agents being studied on animal models, and is already in use in many medical areas. We investigate the short-term effects of PRP and voice rest on voice outcomes following phonosurgery as an alternative to voice rest alone. *Materials and Methods:* A prospective single-blinded pilot study was conducted. Sixteen patients with a diagnosis of vocal fold cyst and polyps were included, forming equal groups (PRP and voice rest vs. voice rest alone). Voice analysis was carried out on the preoperative day, day three, and week three following surgery. The measured parameters were fundamental frequency (F0), noise–signal ratio (NSR), harmonic poverty (HP), attack alteration (AL), pitch instability (PI), and amplitude instability (AI).VHI(Voice Handicap Index)-30 questionnaires were carried out before surgery and three weeks following surgery to assess the impact of subjective voice change on quality of life. PRP was obtained using commercial kits with separator gel. *Results:* An average 3.68-fold increase in platelets was obtained with PRP. No side effects were noted after injection. All voice parameters improved on day three and week three following surgery. Statistical significance was noted only in the fundamental frequency of male patients (*p* = 0.048) in favor of the PRP-voice rest group. In addition, the VHI- 30 questionnaire results between preoperative and postoperative assessments showed statistically significant differences in total VHI score (*p* = 0.02) as well as the physical (*p* = 0.05) and emotional (*p* = 0.02) scale in favor of the PRP-voice rest group. *Conclusions:* PRP presents short term safety in patients who undergo phonosurgery, although long-term outcomes are unknown. PRP and voice rest are superior to voice rest alone when considering subjective assessment of the voice. When analyzing acoustic parameters, PRP and voice rest are not superior to voice rest alone.

## 1. Introduction

Phonosurgery is the primary treatment option for benign vocal fold lesions if primarily-prescribed voice therapy fails [1]. Many postoperative strategies have been described and considered in seeking to minimize the need for voice rest and optimize vocal outcomes. Absolute or relative postoperative voice rest is considered the standard of care for better postoperative functional vocal outcomes. The optimal time and the type of voice rest (absolute, relative) are debatable, ranging from 0 to 14 days in specific cases [2,3,4]. Another matter of concern could be that voice rest as a part of sick leave alone might outweigh the financial burden of the surgery itself [3]. Adjuvant medical treatment options following surgery, such as antibiotics or proton pump inhibitors, do not seem to benefit patients in terms of healing time and voice outcomes [5,6,7].

In light of these limited adjuvant treatment options and the widely discussed optimal voice rest period, intraoperative autologous platelet-rich plasma (PRP) injection may offer benefits if used alone or in conjunction with voice rest. Autologous cellular therapies using PRP are used in all areas of medicine, with particular interest in dermatology, oral surgery, and plastic surgery [8]. At its most basic definition, PRP is a processed fraction of peripheral blood with a platelet concentration above baseline [9]. The rationale for the clinical use of PRP is the release of biologically active factors when applied on an acute injury site, where it can initiate tissue repair through more than 30 bioactive proteins contained in the platelets themselves [9,10].

A few animal studies have reported favorable results in vocal fold healing using PRP following acute vocal fold injury [11,12,13]. To the best of our knowledge, there are currently no human studies regarding the effect of PRP on voice outcomes following phonosurgery. In a recent systematic review, Suresh et al. did not find any clinical studies on PRP and phonosurgery when describing laryngeal applications of PRP [14].

Considering the low side effect profile and the possible benefits, we hypothesized that intraoperative PRP injection following phonosurgery could result in optimal functional outcomes in a shorter period, reducing the need for prologued sick leave.

## 2. Materials and Methods

A prospective randomized single-blinded study was conducted between July 2019 and March 2021. Ethics committee approval from the University of Medicine and Pharmacy “Iuliu Hatieganu” Cluj Napoca and the Emergency County Hospital of Cluj Napoca was granted (41/25.02.2019). Sixteen patients were included in the study (*N* = 16). Patients were assigned to two equal groups: PRP and voice rest (*N* = 8), and voice rest only (*N* = 8). All procedures were carried out at the ENT Department of the Cluj Napoca Emergency County Hospital. All patients participated voluntarily and signed informed consent forms. All procedures in the study followed the norms and ethics of the Helsinki Declaration regarding principles for medical research involving human subjects. The trial was registered on the WHO International Clinical Trials Registry Platform with the TCTR20220205003 registration number.

### 2.1. Selection Criteria

We considered patients eligible if they presented unilateral benign vocal fold pathology requiring phonosurgical intervention and were cleared for PRP preparation. The size of the lesion should not account for more than 50% of the true vocal fold length, and the age should be between 18 and 75 years. Local exclusion criteria were upper respiratory infections within two weeks prior to surgery, clinical signs of malignancy, recurrent respiratory papillomatosis, uncertainty of clinical diagnosis, bilateral vocal fold pathology, prior phonosurgical intervention, voluminous unilateral lesion occupying more than 50% of the length of the vocal fold, sulcus vocalis, and vocal fold paralysis. General exclusion criteria were severe cardiovascular and pulmonary pathology after preanesthesia evaluation, anti-aggregating therapy that could not be interrupted, patients not fit for surgery, patients with known psychiatric conditions, and malignancy. PRP exclusion criteria included thrombocytopenia, platelet disorders, malignancy, hematological disorders, bone marrow aplasia, high white blood count, and sepsis. Using clinical and endoscopic evaluation, two independent otolaryngologists performed patient assessments for inclusion and exclusion. Patients were only included if both examiners agreed on the criteria.

### 2.2. Randomisation Process and Blinding

Patients that met all inclusion criteria and were willing to participate in the study and signed an informed consent form were enrolled in the randomization process. The process was carried out by randomly allocating a pre-defined envelope containing either “PRP” or “NO PRP”. The envelope was opened only during surgery when PRP was to be prepared for the patients in the PRP-voice rest group. As the PRP injection needed to take place, the surgical team could not be blinded through the process. On the other hand, the phoniatrician was blinded, as all participants needed to be evaluated pre- and post-surgically.

### 2.3. Voice Analysis

All patients meeting inclusion criteria underwent voice analysis using GERIP^®^ Vocalab 4 software(LURCO, Toulouse, France), a real time spectral analysis software validated through the French Society for Development of Research in Speech Therapy–UNADREO [15]. Voice analysis was performed three times at three different time points by our departments phoniatrician: one day prior to surgery, three days after surgery, and finally three weeks following surgery. Measured parameters were fundamental frequency (F0), noise–signal ratio (NSR), harmonic poverty (HP), attack alteration (AL), pitch instability (PI), and amplitude instability (AI). The standard threshold values between normal and pathological voice are considered 1.0 for all indicators. Values below 1.0 are considered normal range, and values above 1.0 are considered pathological, as described by the Vocalab software user’s manual.

The recording was performed in a sound-isolated room in the audiology department using a Sennheiser MD-42 omnidirectional microphone (Sennheiser electronic GmbH & Co., KG, Wedemark, Germany) connected to a preamplification system (M Audio^®^ M-Track Plus II) (M-Audio, Cumberland, RI, USA), which was set to 20 cm from the patient’s lips at a 30 degree angle.

Patient were instructed to maintain a relaxed sitting posture with the knees bent and the chin parallel to the floor, keeping a straight posture. Fundamental frequency was determined by measuring the average F0 as the patients spoke for 10 s. All patients were instructed to recite a standard text. Multidimensional voice analysis was carried out by recording a ten-second sustained “a” vowel after a deep inhale.

### 2.4. Voice Hygiene Principles

Voice hygiene concepts were discussed with all patients before admission. Following surgery, voice rest was recommended as absolute until the first postoperative voice assessment (day three) and was continued with relative voice rest for another seven days. During this time, patients were instructed to restrain from any physical activities and habits that can cause glottal attack or strain on the vocal folds.

### 2.5. VHI-30 Questionnaire

On the day of admission and three weeks following surgery a VHI-30 questionnaire was given to each patient to assess the subjective impact of their voice on the quality of life. 

The VHI-30 is a quality-of-life questionnaire validated by Jacobson et al. It evaluates voice disorders’ functional, physical, and emotional aspects. A five-point scale is used for each question (from 0 = never to 4 = always). A maximum of 40 points (10 questions) can be attributed to voice disturbance’s functional, physical, or emotional aspects, with the highest score being 120 for patients with severe voice disorders [16].

### 2.6. Autologous PRP Preparation

For the patients in the PRP and voice rest group, following induction anesthesia, venous blood was drawn in a 12 mL Standard Gel Separation PRP tube (HBH Medical^®^) (HBH Medizintechnik GmbH, Tuttlingen, Germany) under sterile conditions. A centrifuge (model XC-2415) (Merck KGaA, Darmstadt, Germany ) set at 4000 RPM for 5 min was used to separate platelets, as indicated by the provider. The upper two thirds, considered Platelet-Poor Plasma (PPP), was discarded. The lower one third, above the separator gel, was considered PRP (Figure 1). The PRP was aspirated in a 2 mL syringe, and a 26 G spinal needle(Shanghai Mekon Medical Devices Co. Ltd., Shanghai, China) was attached for delivery. Before injection, a fraction of the separated PRP was analyzed for the platelet count to ensure that the product met the minimal platelet concentration requirements to be considered PRP.

### 2.7. Surgical Procedure

Using cold steel instruments, the standard surgical procedure was suspension microlaryngoscopy (MLS) under microscopic guidance (Figure 2). After the lesion was ablated and hemostasis was obtained, 1.5 mL of PRP was injected into the upper surface of the vocal fold just lateral to the surgical area until slight medialization was obtained. Delivery was carried out through a 26 G 90 mm spinal needle attached to a 2 mL syringe. A single dose of 1000/200 mg of Amoxicillin and Clavulanic acid was administered intraoperatively. No treatment except proton pump inhibitors was prescribed following surgery for all patients.

### 2.8. Statistical Analysis

Statistical analysis was performed using SPSS software v.25 (SPSS Inc., Chicago, IL, USA); descriptive statistics are presented as mean ± standard deviation. A *t*-test was applied to evaluate population demographics and acoustic parameters and to assess group homogeneity. Category variables were calculated using a Chi-squared test. Repeated measures ANOVA was used to evaluate the voice parameters and VHI questionnaire results (two groups at three time intervals). Statistical significance was considered at *p* < 0.05.

## 3. Results

Sixteen patients with benign vocal fold pathology meeting all inclusion criteria were evaluated. Eight patients received intraoperative PRP injections, and eight patients were enrolled as control. Population characteristics are shown in Table 1. No statistically significant characteristics were observed between the two groups, resulting in a homogenous population.

An average of a 3.68-fold increase of platelets was obtained in PRP compared to venous blood, which is in line with the minimal platelet concentration requirement for PRP. Detailed results are shown in Table 2.

There were no statistically significant differences between groups regarding the preoperative variables, except for F0 in the VR group, where F0 was significant in females (*p* = 0.007).

All vocal parameters in both groups showed improvement from the initial preoperative measurement. Statistical significance between the groups can be observed in the fundamental frequency of the male patience (*p* = 0.048) in favor of the PRP and voice rest group. Results are shown in Table 3.

Repeated measure ANOVA was applied for voice parameters in smokers and non-smokers in male and female patients, with no statistical differences observed (*p* > 0.05).

The VHI-30 questionnaire results between preoperative and postoperative assessments show statistically significant differences on the physical (*p* = 0.05) and emotional (*p* = 0.02) scales. Both groups averaged lower scores on the postoperative evaluations, with a better improvement in the PRP-voice rest group. Although the functional scale was not statistically significant (*p* = 0.25) between the groups, a great improvement was noted in all patients. Total VHI score decreased significantly more in PRP-VR group at three weeks after surgery compared to the VR group (*p* = 0.02). A larger study population would probably result in statistical significance. Regarding the preoperative measurements, only the emotional subscale values were significantly higher in the PRP-VR group as compared with the VR group (*p* = 0.01).(Table 4)

## 4. Discussion

Improvement and long-term optimal voice maintenance must be the major focus and outcome of phonosurgery. Considering that the surgery itself is an acute traumatic event, preservation of the normal architecture and surface anatomy in the early stage of healing is of utmost importance. In the last decades, no improvements have been made regarding adjuvant treatment options to prevent chronic scarring. While voice rest is considered the standard of care in the immediate postoperative period, there are contradictory results regarding its real benefits for the patient.

First, it is unclear if voice rest is needed following phonosurgery. Second, if voice rest is recommended, the optimal period is unclear. In a recent study, G. Bjork et al. found no differences in 488 patients regarding VHI-10 scores at four months between postoperative phonation and non-phonation groups [17]. On the other hand, D. Kiagiadaki et al. found that a ten-day voice rest period yielded better voice outcomes [4]. The problem of chronic scarring has not been adequately addressed due to the lack of treatment options [18].

Researchers have been looking for alternative adjuvant treatment options, including growth factors, hyaluronic acid, and stem cells in recent years. However, most of the vocal fold healing and scarring data, including novel treatment options, have been in the phases of fundamental or animal research [19,20,21,22]. Transitioning to clinical trials is confined by regulations, safety, and ethical issues. Thus, implementation takes more time and effort. 

PRP is an autologous product that is used in numerous areas of medicine. Although the lack of standardization and different classifications of PRP make it hard to compare results and ultimately evaluate the product’s utility, the safety profile does not seem to raise any concerns.

To the best of our knowledge, this is the first study to evaluate the functional effects of PRP on voice outcomes following phonosurgery in human subjects. This prospective randomized (single-sided) pilot study aimed to substantiate whether intraoperative vocal fold PRP injections could contribute to better voice outcomes in patients undergoing phonosurgery and to assess whether PRP can represent a viable adjuvant treatment option in the future. Being a pilot study, we wanted to establish risk, benefit, workflow, evaluation protocol, and the opportunity for a larger group distribution with subsequent protocol enhancements.

Assuming the complexity of vocal fold pathology, anatomy, and the variable nature of patient comorbidities and habits, we adhered to a narrow inclusion protocol to avoid bias. Ultimately, the inclusion yielded only patients with vocal fold polyps and cysts. We assumed equal groups, a standardized surgical procedure, strict voice analysis, and subjective assessment of voice impact. Statistical analysis showed homogenous study groups.

Currently, several methods of acquiring PRP exist. Earlier methods used single or double centrifugation methods and manual buffy coat separation. Nowadays, commercially available automated kits with or without separator gel are more popular and have more regulatory processes, making them a safe option for daily use. In our opinion, manual methods are no longer suitable due to lack of standardization and the high probability of sample contamination. Commercial kits are safe alternatives due to the closed nature of platelet separation. Our study used commercial kits with separator gel to ensure adequate platelet separation and concentration. This type of PRP, referring to the classification of Ehrenfest et al. [23], can be categorized as L-PRP, a product containing both platelets and white blood cells. We obtained a 3.68 ± 0.39-fold increase in platelets compared to whole blood, which qualifies as PRP as defined by two classification systems [24,25].

The voice analysis evaluated six parameters for voice disturbances. Results came in the form of graphs and numerical values for each parameter. Interpretation values for all parameters were considered normal below 1.0. For fundamental frequency (F0), we split the results for male and female voices due to the significant difference in normal values. The average healthy fundamental frequency for men is reported at 115 Hz, while for women the average is reported as being around 210 Hz [26,27,28]

We found no statistically significant differences between the two groups regarding attack alteration, pitch instability, amplitude instability, harmonic poverty, or signal to noise ratio. We observed significant improvement on day three and postoperative week three in both groups. At postoperative week three, attack alteration, pitch instability, and amplitude instability showed values slightly above 1.0 or under 1.0, which is considered normal as interpreted by the Vocalab^®^ software. Harmonic poverty (PRP-voice rest =1.43 vs. voice rest = 1.70) and noise to signal ratio (PRP-voice rest = 1.64 vs. voice rest = 1.50), although improved, remained in the pathological interval. 

For fundamental frequency, we obtained statistical significance for the male voices on day three and week three, with better improvement in the PRP-voice rest group, where preoperative analysis averaged 150 Hz F0, 127.5 Hz on day three, and 109.75 Hz at three weeks. Postoperative values are close to the average of 115 Hz found in the literature. Women’s F0 did not suffer significant modifications from the preoperative measurements to the three day and three-week measurements.

Subjective assessment using the VHI-30 questionnaire yielded statistically significant differences between the two groups on the total, physical, and emotional scales. Better results in favor of the PRP-voice rest group were noted. The fact that the study was single-blinded gives the VHI-30 results greater reliability and weight in comparison to the voice analysis, where vocal parameters were not statistically significant between the two groups. 

We report no short-term side effects or prolonged inflammation at the site of the PRP injections. All procedures were well tolerated. We believe that the presented results are promising and that further investigations regarding the use of PRP injections following phonosurgery could be beneficial. As a result of this pilot study, we would conclude a series of benefits and shortcomings.

As we did not observe less favorable outcomes or complications in the PRP-VR compared to VR alone, we can further support the safe nature of PRP for short-term use. Due to the safety profile, even partial or minor improvements can be considered beneficial due to the risk–benefit ratio. Follow up on patients at six months would bring valuable information and could further consolidate the short-term findings.

PRP-VR is more effective than VR alone when evaluating results through the VHI-30 questionnaire. While postoperative improvements could be noted in both groups, better outcomes were seen in the VR-PRP group in the total, physical, and emotional scores. 

On the other hand, voice analysis did not show any statistical significance between the two groups. The fact that there was a statistical difference in F0 in favor of the male patients in the PRP-VR group might be because of the limited number of patients. 

Although only partial results showed statistical significance, we believe that a larger population could bring further contribution. This shortcoming can be considered one of the major limitations of the study. Another limitation is the suboptimal performance of the microphone, which did not meet all the requirements described by Patel et al. [29].

As to future directions, we believe that more clinical trials are needed with larger study populations and that different types of PRP and different platelet concentrations must be taken into consideration as well. Different PRP products might yield different results, and should be evaluated. P-PRP (pure PRP, that is, PRP with no leucocytes) could hypothetically be a better alternative, as the lack of white blood cells could lead to a more moderate inflammatory phase response. Considering the fact that we were conservative, not knowing the potential outcomes, and compared PRP and voice rest to voice rest alone, in the future it would be more beneficial to compare the effect of PRP alone to voice rest in order to investigate whether it can positively influence sick leave, voice parameters, and quality of life.

## 5. Conclusions

PRP is safe to use in the short term as an adjuvant treatment option in patients who undergo phonosurgery, although long-term effects need to be evaluated as well. PRP and voice rest are superior to voice rest alone when considering the subjective assessment of patient voice (VHI-30 questionnaire). No statistically significant differences were noted in voice parameters between the two groups; when considering acoustic parameters, PRP and voice rest were not superior to voice rest alone. More evidence is needed to support using PRP alone without voice rest as a treatment option for post-surgical rehabilitation.

## Figures and Tables

**Figure 1 medicina-58-00988-f001:**
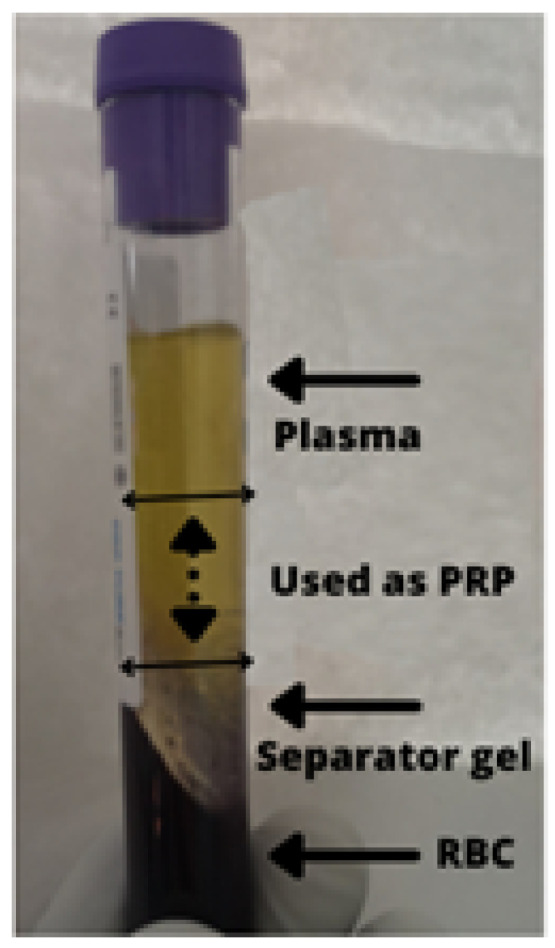
Fractional separation of PRP using gel separator tubes: the plasma or platelet-poor plasma fraction was discarded. The injected fraction or PRP is outlined with an interrupted arrow. Approximately 2 mL was obtained from which 1.5 mL was injected in the vocal fold and 0.5 mL was sent for platelet count. PRP = platelet rich plasma; RBC = red blood cells.

**Figure 2 medicina-58-00988-f002:**
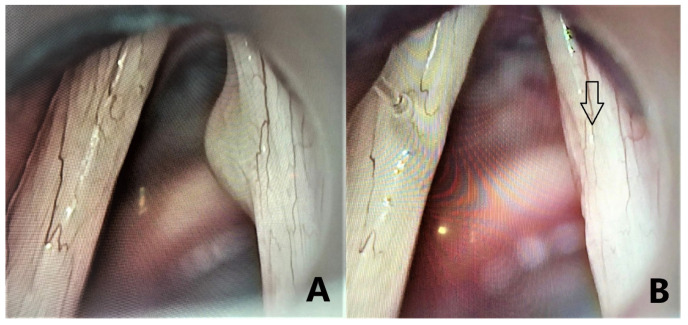
Suspended laryngoscopy with vocal fold cyst excision: (**A**) preoperative image of right vocal fold cyst and (**B**) postoperative image after hemostasis was obtained. The site of the injection is marked with an arrow.

**Table 1 medicina-58-00988-t001:** Group Characteristics.

Grup Characteristics	PRP (*N* = 8)	Non-PRP (*N* = 8)	*p* Value
Age	51.25 ± 8598	54.25 ± 13,823	0.61 ^1^
Sex (m/f)	3(37.5%)/5(62.5%)	4(50%)/4(50%)	1.00 ^2^
Pathology (c/p)	2(25%)/6(75%)	3(37.5%)/5(62.5%)	1.00 ^2^
Smoking(s/ns)	5(62.5%)/3(37.5%)	4(50%)/4(50%)	1.00 ^2^

m = male; f = female; c = vocal fold cyst; p = vocal fold polyp; s = smoker; ns = non-smoker; ^1^ = *t* test; ^2^ = chi square test.

**Table 2 medicina-58-00988-t002:** Individual PRP characteristics in PRP-voice rest group.

Nr	Sex(M/F)	Platelets (10^9^/L)	PRP (10^9^/L)	Fold Increase
1	M	393	1291	3.28
2	F	162	593	3.66
3	M	196	779	3.97
4	M	224	831	3.71
5	F	310	1251	4.04
6	M	317	1041	3.28
7	F	204	874	4.28
8	F	278	893	3.21
**Average ± St.deviation**		**260.50 ± 76.36**	**944.13 ± 237.47**	**3.68 ± 0.39**

Data representing the average platelet count (109/L) from the whole blood and platelet count (109/L) after PRP separation. Values only represents patients in the PRP group. M = male; F = female.

**Table 3 medicina-58-00988-t003:** Acoustic analysis comparison between PRP-voice rest and voice rest groups VR = voice rest group.

Variables	Groups		Preoperative			3 Days Postoperative		3 Weeks Postoperative		*p* **
**Attack alteration**			**Mean**	**SD**	** *p* ** *****	**Mean**	**SD**	**Mean**	**SD**	**0.313 ^1^**
	VR		2.16	±0.63	0.398	1.56	±0.25	0.97	±0.16	
	PRP-VR		2.44	±0.65		1.46	±0.35	0.90	±0.13	
**F0**	VR	M	124.00	±20.94	0.007	123.60	±13.50	111.00	±7.64	**0.048 ^1^**
		F	181.67	±17.01		181.00	±7.55	199.33	±6.65	**0.263 ^1^**
	PRP-VR	M	150.00	±21.43	0.055	127.50	±5.91	109.75	±4.57	**0.048 ^1^**
		F	207.00	±43.01		207.00	±27.94	198.00	±9.27	**0.263 ^1^**
**Pitch Instability**	VR		2.31	±0.60	0.474	1.68	±0.44	1.17	±0.36	**0.201 ^1^**
	PRP-VR		2.55	±0.68		1.52	±0.74	1.10	±0.35	
**Amplitude instability**	VR		1.59	±0.41	0.173	1.27	±0.26	0.89	±0.28	**0.832 ^1^**
	PRP-VR		1.92	±0.49		1.5	±0.61	1.07	±0.30	
**Harmonic** **poverty**	VR		3.20	±0.90	0.926	2.41	±0.33	1.70	±0.76	**0.907 ^1^**
	PRP-VR		2.97	±1.45		2.37	±1.06	1.43	±0.89	
**NSR**	VR		3.00	±1.11	0.703	2.28	±0.87	1.50	±0.58	**0.964 ^1^**
	PRP-VR		3.24	±1.33		2.43	±1.03	1.64	±0.68	

PRP-VR = PRP and voice rest group; * *p* value for preoperative comparison; ** *p* value for repeated measurements; F0 = fundamental frequency; NSR = noise to signal ratio; ^1^ = repeated measures ANOVA.

**Table 4 medicina-58-00988-t004:** Subjective voice assessment between groups using the VHI-30 questionnaire.

VHI Subscale	Groups	Preoperative		3 Weeks Postoperative		*p* Value
**Functional**		Mean	Std. Deviation	Mean	Std. Deviation	**0.25 ^1^**
	VR	14.00	±4.37	5.62	±1.68	
	PRP-VR	15.37	±6.18	3.75	±2.12	
**Physical**	VR	17.50	±2.44	6.37	±1.50	**0.05 ^1^**
	PRP-VR	20.87	±6.22	4.50	±2.44	
**Emotional**	VR	9.00	±3.25	5.37	±1.50	**0.02 ^1^**
	PRP-VR	14.87	±4.79	3.62	±2.97	
	VR	40.5	±6.45	17.37	±4.03	**0.02 ^1^**
	PRP-VR	51.12	±15.59	11.87	±6.99	

The maximum and minimum scores for each subsection (functional, physical, emotional) can be seen in the methods section; VR = voice rest group; PRP-VR = PRP and voice rest group; ^1^ = repeated measures ANOVA; VHI = voice handicap index.

## Data Availability

The data presented in this study are available on request from the corresponding author. The data are not publicly available due to privacy and ethical reasons.

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
