# Peer review of "The Effect of Platelet-Rich Plasma Injection on Short Term Vocal Outcomes Following Phonosurgery—A Pilot Study"

_medicina, 2022, doi:10.3390/medicina58080988_

Round 1
Reviewer 1 Report
It is an interesting paper on the use of platelet rich plasma in phonosurgery. There are some minor deficiencies which should be improved.
Figure 1. All the abbreviations used in the figure should be explained.
The authors should give information on the duration of voice rest – 3 weeks? Complete voice rest, relative voice rest?
There is a considerable difference in F0 between the patients with VR and VR-PRP. The autors should at least discuss it.
Line 210: The authors write, »Repeated measure ANOVA was also applied for smokers and non-smokers in male and female patients, with no statistical differences observed (p>0.05)«. They should explain what was compared (results of acoustic analysis?, VHI 30?)
There are some typing errors (e.g. lines 27, 186, 191).
Author Response
Comment: Figure 1. All the abbreviations used in the figure should be explained.
Response: Thank you for bringing this to our attention. We have made the appropriate changes
Comment: The authors should give information on the duration of voice rest – 3 weeks? Complete voice rest, relative voice rest?
Response: Thank you. We have addressed this issue in the Methods section where we explain the voice rest period and the concept of vocal hygiene used by the patience.
Comment: There is a considerable difference in F0 between the patients with VR and VR-PRP. The autors should at least discuss it.
Response: We appreciate the comment. We have added more information as we extended the statistics with the total VHI score as well.
Comment: Line 210: The authors write, »Repeated measure ANOVA was also applied for smokers and non-smokers in male and female patients, with no statistical differences observed (p>0.05)«. They should explain what was compared (results of acoustic analysis?, VHI 30?)
Response: We appreciate bringing this to our attention. We were referring to voice analysis and added this point to the main text as well.

Reviewer 2 Report
In this paper, the authors evaluated in a single-blind RCT various voice outcomes (acoustics and self-evaluation) of two therapy methods (platelet rich plasma plus voice rest (PRP-VR) vs. voice rest (VR) alone) for patients with vocal fold cysts and polyps. It is a valuable topic and interesting paper. However, I have some remarks, which leads to my conclusion of a major revision.
Major comments:
· Material and Methods: Line 65-69. The randomization process is unclear. Please provide here further information on how and by whom the randomization was organized.
· Material and Methods: Line 65-69. The blinding process is unclear. Please explain detailly who was blinded (e.g., patients of the therapy groups [if yes, how was it controlled and organized], examiners of the acoustics, etc.).
· Abstract. Line 31-32. A superior effect for PRP-VR cannot be mentioned here. Just two from four parameters of the VHI showed a slight significant effect (p<= 0.05). The functional component is above and the total score (most important in literature) is missing.
· Abstract. Line 32-34. This paragraph is confusing. First, why is it important to have 3 to 4 Hz lower F0 than the VR-group. Furthermore, the start conditions seems uneven (VR-group pre: 124 Hz vs PRP-VR pre: 150 Hz with comparable SD in both groups). Second, to highlight just one acoustic parameter of a batterie of six parameters is a distortion and doesn’t represent the results. To avoid this kind of distortion a Bonferroni correction of the p-values is necessary and you would receive no “superior” effect of PRP-VR in the present study.
· Material and method. It is unclear if voice rest is meant by only prohibition of voice use or included also other functions such as coughing, laughing, lifting heavy objects etc. Furthermore, what kind of voice rest was used (absolute or relative)? In addition, was a vocal hygiene concept discussed with patients or only limited on voice rest? Finally, how was voice rest controlled and for how many days was an absolute or relative voice rest ordered after the surgery? Please add here more information about this topic.
· Material and Method. Line 113-114. A total score is missing in the whole manuscript. Please adjust here and in the result section the scores of the total value with all necessary statistics in Table 4.
· Voice analysis. Line 104-105. Method: The recording system of data acquisition is unknown. There are guidelines of technical data for microphones for voice and speech analyses such as by Patel et al. (2018) [Patel RR, Awan SN, Barkmeier-Kraemer J, et al. Recommended protocols for instrumental assessment of voice: American Speech-Language-Hearing Association expert panel to develop a protocol for instrumental assessment of vocal function. American Journal of Speech-Language Pathology. 2018;27(3):887-905.]. Thus, it is dubious how valid the results of the present study are. Please add here more information and hopefully the hardware fulfill the criteria.
·
· Statistics. Line 148-153. There are no statistics to investigate the start conditions (preoperative) of the voice parameters between the two groups. There are doubts that these results are equal in all parameters (e.g., male F0, and VHI-total[is missing parameter])
· The authors are advised to be more conservative in their recommendation on the PRP-VR therapy method for the whole manuscript because, firstly, the p-values are not highly significant (p< 0.01) and, secondly, most of the voice parameters from the diagnostic battery did not show a significant effect in the ANOVA repeated-measures. Based on the current data, fact seems to be that VR alone is a sufficient rehabilitation method after phonosurgery. To be clarified and missing as a third arm of the study is the effect of voice treatment by a speech-language pathologist and/or vocal hygiene.
Minor comments:
· Abstract: In the material and method section the voice outcomes for acoustics is missing. Please add here the necessary information such as in the main text.
· Abstract: Please adjust the following text “in favor of the PRP and voice rest group” into “in favor of the PRP-voice rest group”
· Abstract: Please adjust the following text “in favor of the PRP group” into “in favor of the PRP-voice rest group”
· Abstract: Please adjust the following text “long-term outcomes are not known” into “long-term outcomes are unknown”
· Introduction: Please add more information about PRP therapy such as, how often is it used in rehabilitation during or phonosurgery, chances of healing success and its precise results.
· Voice analysis: Line 95. What is precisely the GERIP Vocalab software, because it is not a gold standard in acoustic voice analyses such as MDVP or Praat? Please add here more information and references.
· Voice analysis: 100-103. It is unclear how the norm values were created. Are there clinical studies which validated the software? If not, it is part of bias and has to be mentioned in the limitations of the present study.
· Discussion. At the end of the discussion section a limitation paragraph is missing and organize much better future directions. Please add these information.
Author Response
Major Comments:
Comment: Material and Methods: Line 65-69. The randomization process is unclear. Please provide here further information on how and by whom the randomization was organized. Material and Methods: Line 65-69. The blinding process is unclear. Please explain detailly who was blinded (e.g., patients of the therapy groups [if yes, how was it controlled and organized], examiners of the acoustics, etc.).
Response: You are right, and we appreciate your recommendation. We added a new subsection to methods (2.2) which describes the randomization and the blinding process as well.
Comment: Abstract. Line 31-32. A superior effect for PRP-VR cannot be mentioned here. Just two from four parameters of the VHI showed a slight significant effect (p<= 0.05). The functional component is above and the total score (most important in literature) is missing. Abstract. Line 32-34. This paragraph is confusing. First, why is it important to have 3 to 4 Hz lower F0 than the VR-group. Furthermore, the start conditions seems uneven (VR-group pre: 124 Hz vs PRP-VR pre: 150 Hz with comparable SD in both groups). Second, to highlight just one acoustic parameter of a batterie of six parameters is a distortion and doesn’t represent the results. To avoid this kind of distortion a Bonferroni correction of the p-values is necessary and you would receive no “superior” effect of PRP-VR in the present study.
Response: Thank you for pointing this out. We have adjusted the abstract accordingly and made modifications in the main text to be more withheld with the comments regarding the results. Also, we believe the Bonferroni correction it is not applicable here, as we are not conducting multiple analyses on the same dependent variable
Comment: Material and method. It is unclear if voice rest is meant by only prohibition of voice use or included also other functions such as coughing, laughing, lifting heavy objects etc. Furthermore, what kind of voice rest was used (absolute or relative)? In addition, was a vocal hygiene concept discussed with patients or only limited on voice rest? Finally, how was voice rest controlled and for how many days was an absolute or relative voice rest ordered after the surgery? Please add here more information about this topic.
Response: Thank you for the valuable input. Indeed, more information was needed regarding how the voice rest and voice hygiene concept was applied. We added a new paragraph to the Methos, detailing this information.
Comment: Material and Method. Line 113-114. A total score is missing in the whole manuscript. Please adjust here and in the result section the scores of the total value with all necessary statistics in Table 4.
Response: Thank you so much for bringing this to our attention. It is a big miss from our part. We added total values in all the manuscript and changed the statistics, table, and paragraphs accordingly.
Comment: Voice analysis. Line 104-105. Method: The recording system of data acquisition is unknown. There are guidelines of technical data for microphones for voice and speech analyses such as by Patel et al. (2018) [Patel RR, Awan SN, Barkmeier-Kraemer J, et al. Recommended protocols for instrumental assessment of voice: American Speech-Language-Hearing Association expert panel to develop a protocol for instrumental assessment of vocal function. American Journal of Speech-Language Pathology. 2018;27(3):887-905.]. Thus, it is dubious how valid the results of the present study are. Please add here more information and hopefully the hardware fulfill the criteria.
Response: We appreciate the remark and added information regarding the hardware system, microphone and preamp. All components of the setup meet minimum requirements as according to Patel et al.
Comment: Material and Method. Line 113-114. A total VHI score is missing in the whole manuscript. Please adjust here and in the result section the scores of the total value with all necessary statistics in Table 4.
Response: Thank you for the suggestions. We added the necessary data (line 246-249 and the last line from table 4).
Comment: Statistics. Line 148-153. There are no statistics to investigate the start conditions (preoperative) of the voice parameters between the two groups. There are doubts that these results are equal in all parameters (e.g., male F0, and VHI-total[is missing parameter])
Response:Thank you for the suggestion. Although we believe it is more important how these variables changed in time, we followed your instructions. We added the requested information
Minor Comments:
Comment: Abstract: In the material and method section the voice outcomes for acoustics is missing. Please add here the necessary information such as in the main text.
Response: Thank you for pointing his out. We added the parameter values of the acoustic measures
Comment: Abstract: Please adjust the following text “in favor of the PRP and voice rest group” into “in favor of the PRP-voice rest group”; Please adjust the following text “in favor of the PRP group” into “in favor of the PRP-voice rest group”; Please adjust the following text “long-term outcomes are not known” into “long-term outcomes are unknown”
Response: Thank you, we have made the appropriate adjustments
Comment: Introduction: Please add more information about PRP therapy such as, how often is it used in rehabilitation during or phonosurgery, chances of healing success and its precise results.
Response: We appreciate the comment. To the best of our knowledge there are no studies investigating the effect of PRP on rehabilitation during phonosurgery. We further included a recent systematic review that investigates laryngeal application of PRP but no articles related to phonosurgery were included either.
Comment: What is precisely the GERIP Vocalab software, because it is not a gold standard in acoustic voice analyses such as MDVP or Praat? Please add here more information and references. It is unclear how the norm values were created. Are there clinical studies which validated the software? If not, it is part of bias and has to be mentioned in the limitations of the present study.
Response: Thank you for the input. GERIP is a software developed in France and was validated by the French Society for Development of Research in Speech Therapy – UNADREO. We added information about this in the methods section. Unfortunately, this is the software what the University provided us together with the hardware. Thank you.
Comment: The authors are advised to be more conservative in their recommendation on the PRP-VR therapy method for the whole manuscript because, firstly, the p-values are not highly significant (p< 0.01) and, secondly, most of the voice parameters from the diagnostic battery did not show a significant effect in the ANOVA repeated-measures. Based on the current data, fact seems to be that VR alone is a sufficient rehabilitation method after phonosurgery. To be clarified and missing as a third arm of the study is the effect of voice treatment by a speech-language pathologist and/or vocal hygiene.
Response: Thank you for the comment. As stated even from the title we consider this a pilot study with just a few enrolled patients. Unfortunately, the pandemic did not allow us to enroll more patients in the study so with such a small groups distribution a p<0.01 would be a stretch. The text has been revised and more conservative recommendations have been adopted. A third arm of the study, we believe would not bring a benefit because we wanted to see if PRP brings any positive addition to the current standard treatment option, secondly speech therapy would not yield any satisfactory results in vocal fold cysts and polyps (no true feedback and face to face limitations due to the pandemic). Further, patients were recommended surgery due to the lack of response from voice therapy.

Round 2
Reviewer 2 Report
Thanks to the authors for the adjustments to their previous version. I greatly appreciate the authors' efforts to address my comments and strengthen the manuscript. However, a close reading reveals three points to which I would like to draw attention and ask the authors to work on this precisely, carefully and comprehensively before the manuscript can be released from my point of view:
1) I have significant doubts that the hardware for acoustic voice analysis meets the requirements as described e.g. in Patel et al. (2018). Firstly, Sennheiser MD-42 is not a head-mounted microphone, which is important from two aspects: (1) ensure always the same distance, (2) reduces environmental noise, which have a critical impact on the validity of the outcomes of the acoustic measures. Secondly, the distance should be between 4-10 cm but the present study used 20 cm. Thirdly, the angle of the position of the microphone is still unknown.
In addition, I had a look at the manual of the MD-42 (https://assets.sennheiser.com/global-downloads/file/783/MD42_bda.pdf). The most important parameters of the hardware were not met, as described under the paragraph of the technical specifications of the paper from Patel et al. (2018), which was also mentioned in the reference list from the authors. First, a flat frequency response (i.e., variation of less than 2 dB) across the frequency range between the lowest expected F0 of voice and the highest spectral component of interest (approximately 50–8000 Hz). The following picture is copied from the MD42 manual, in which a peak of about 10 dB are registered of the critical range (see image below).
Second, the equivalent noise level is still unclear because it is not described in the manual. Third, the dynamic range with its maximum intensity (i.e., measured with 3% THD) is also not mentioned in the manual.
Next to the microphone information there are missing information about the digital recording as described in the eponymous named section of the paper from Patel et al. (2018) as well.
All in all, it is not appropriate to write about “hardware setup meets the minimum requirements as described by Patel et al.”. I disagree with this author's statement, and in consequence the measures of acoustic analyses may introduce a bias in the results that needs to be discussed in the limitations of the study as well. Thus, please adjust and consider all remarks from the list as mentioned above in your method and discussion sections.
A final remark: The authors would be well advised to use a different recording technique for their future studies and to pay more attention to the standards of acoustic analysis.
2) Line 132 to 136. The new paragraph does not make sense under the sub headline 2.3 voice analysis. Please adjust.
3) Line 286-287. This new passage informs the reader of no significant differences in the preoperative variables. It is solved by the authors very minimalistic and on top it is placed in the wrong place in the manuscript (i.e., after the description of the pre-post results). I am missing information which test was used (information in the statistical section of 2.7.), and which outcomes exactly were measured to assess equal conditions between groups (please adjust here table 3). Again, I have still doubts that the F0 values are preoperatively equal in the male groups.

Author Response
Response: We really appreciate the comment and checking the microphone data sheet; indeed, some technical specifications are missing or do not meet all recommendations as cited. Thank you for pointing it out. Once the equipment was acquired for acoustic analysis, we did not think that it would pose any problems, it is something that we should have checked. Moving forward we will upgrade the recording system. Ultimately, we have modified and moved and the paragraph “hardware setup meets the minimum requirements as described by Patel et al.” to the Discussions section and noted it as a limitation of the study (line 352-354).
Comment: Line 132 to 136. The new paragraph does not make sense under the sub headline 2.3 voice analysis. Please adjust.
Response: Thank you. We added a new subsection (2.4) discussing only the voice hygiene.
Comment: Line 286-287. This new passage informs the reader of no significant differences in the preoperative variables. It is solved by the authors very minimalistic and on top it is placed in the wrong place in the manuscript (i.e., after the description of the pre-post results). I am missing information which test was used (information in the statistical section of 2.7.), and which outcomes exactly were measured to assess equal conditions between groups (please adjust here table 3). Again, I have still doubts that the F0 values are preoperatively equal in the male groups.
Response: Thank you for pointing this out. We have added the text for statistical analysis of acoustic parameters in the statistical analysis (line 184). We adjusted the lines 227-229 for preoperative variable significance. Also a new column was added for Table 3 with the P values of preoperative variables.
